# A Dedicated Veno-Venous Extracorporeal Membrane Oxygenation Unit during a Respiratory Pandemic: Lessons Learned from COVID-19 Part I: System Planning and Care Teams

**DOI:** 10.3390/membranes11040258

**Published:** 2021-04-02

**Authors:** Sagar Dave, Aakash Shah, Samuel Galvagno, Kristen George, Ashley R. Menne, Daniel J. Haase, Brian McCormick, Raymond Rector, Siamak Dahi, Ronson J. Madathil, Kristopher B. Deatrick, Mehrdad Ghoreishi, James S. Gammie, David J. Kaczorowski, Thomas M. Scalea, Jay Menaker, Daniel Herr, Eric Krause, Ali Tabatabai

**Affiliations:** 1Department of Surgery, Program in Trauma, School of Medicine, University of Maryland, Baltimore, MD 21201, USA; sagar.dave@umm.edu (S.D.); kgeorge@umm.edu (K.G.); tscalea@som.umaryland.edu (T.M.S.); dherr@som.umaryland.edu (D.H.); 2Department of Surgery, Division of Cardiac Surgery, School of Medicine, University of Maryland, Baltimore, MD 21201, USA; sdahi@som.umaryland.edu (S.D.); rmadathil@som.umaryland.edu (R.J.M.); kdeatrick@som.umaryland.edu (K.B.D.); mghoreishi@som.umaryland.edu (M.G.); jgammie@som.umaryland.edu (J.S.G.); 3Department of Anesthesiology, Program in Trauma, School of Medicine, University of Maryland, Baltimore, MD 21201, USA; sgalvagno@som.umaryland.edu; 4Department of Emergency Medicine, Program in Trauma, School of Medicine, University of Maryland, Baltimore, MD 21201, USA; amenne@som.umaryland.edu (A.R.M.); dhaase@som.umaryland.edu (D.J.H.); 5Perfusion Services, University of Maryland Medical Center, Baltimore, MD 21201, USA; bmccormi@umm.edu (B.M.); rrector@umm.edu (R.R.); 6Department of Cardiothoracic Surgery, University of Pittsburgh Medical Center, Pittsburgh, PA 15213, USA; kaczorowskidj2@upmc.edu; 7Department of Surgery, University of California San Francisco Medical Center, San Francisco, CA 94143, USA; jay.menaker@ucsf.edu; 8Department of Surgery, Division of Thoracic Surgery, School of Medicine, University of Maryland, Baltimore, MD 21201, USA; ekrause@som.umaryland.edu; 9Department of Medicine, Division of Pulmonary and Critical Care, Program in Trauma, School of Medicine, University of Maryland, Baltimore, MD 21201, USA; atabatabai@som.umaryland.edu

**Keywords:** extracorporeal membrane oxygenation, COVID-19, acute respiratory distress syndrome, biocontainment unit

## Abstract

Background: The most critically ill patients with coronavirus disease 2019 (COVID-19) may require advanced support modalities, such as veno-venous extracorporeal membrane oxygenation (VV-ECMO). A systematic, methodical approach to a respiratory pandemic on a state and institutional level is critical. Methods: We conducted retrospective review of our institutional response to the COVID-19 pandemic, focusing on the creation of a dedicated airlock biocontainment unit (BCU) to treat patients with refractory COVID-19 acute respiratory distress syndrome (CARDS). Data were collected through conversations with staff on varying levels in the BCU, those leading the effort to make the BCU and hospital incident command system, email communications regarding logistic changes being implemented, and a review of COVID-19 patient census at our institution from March through June 2020. Results: Over 2100 patients were successfully admitted to system hospitals; 29% of these patients required critical care. The response to this respiratory pandemic augmented intensive care physician staffing, created a 70-member nursing team, and increased the extracorporeal membrane oxygenation (ECMO) capability by nearly 200%. During this time period, 40 COVID-19 patients on VV-ECMO were managed in the BCU. Challenges in an airlock unit included communication, scarcity of resources, double-bunking, and maintaining routine care. Conclusions: Preparing for a surge of critically ill patients during a pandemic can be a daunting task. The implementation of a coordinated, system-level approach can help with the allocation of resources as needed. Focusing on established strengths of hospitals within the system can guide triage based on individual patient needs. The management of ECMO patients is still a specialty care, and a systematic and hospital based approach requiring an ECMO team composed of multiple experienced individuals is paramount during a respiratory viral pandemic.

## 1. Introduction

Since the creation of the 1918 Empyema Commission in response to the Spanish Flu that claimed over 50 million lives worldwide, the importance of a coordinated clinical response to a viral pandemic has been recognized [1]. More than 100 years later, the novel coronavirus disease 2019 (COVID-19) threatens the world with a similar death toll [2]. A system-based response is vital to facing disasters such as a viral pandemic by increasing situational awareness and team reliability [3,4]. In 2014, our institution readied itself for the possible outbreak of Ebola in the United States. At that time, our institution set up a system-wide and hospital incident command system (HICS) and a biocontainment unit (BCU). While we never fully realized these plans for Ebola, the experience became the basis for our early response to COVID-19. 

However, unlike Ebola, COVID-19 quickly became a global pandemic. Along with the other hospital systems worldwide, we had to quickly adapt to shortages in staffing, bed availability, medications, and overall lack of understanding of pathophysiology. With reports of extracorporeal membrane oxygenation (ECMO) use in the most severely affected patients, the interim 2019 Extracorporeal Life Support Organization report advised centers with ECMO capabilities to responsibly allocate resources and advised against offering ECMO support in centers with limited experience and/or critical care resources. Keeping in line with these recommendations, whilst increasing overall critical care capacity for COVID-19 patients, we sought to offer ECMO to as many patients as possible given our extensive experience with this support modality in a dedicated lung rescue unit (LRU) [5,6,7,8]. A BCU was created to treat patients with refractory COVID-19 acute respiratory distress syndrome (CARDS) that would require veno-venous (VV) ECMO. This paper describes our approach to the creation of a dedicated unit that cared for 40 VV-ECMO patients during a respiratory viral pandemic, and the lessons learned from our experience.

## 2. Materials and Methods

We conducted retrospective review of our institutional response to the COVID-19 pandemic, focusing on the creation of a dedicated BCU created to treat patients with refractory CARDS. Institutional review board approval was obtained and need for consent was waived for this study (HP-00090914). Data were collected through conversations with staff on varying levels in the BCU, those leading the effort to make the BCU and HICS, email communications regarding logistic changes being implemented, and a review of COVID-19 patient census at our institution from March through June Descriptive statistical analysis was performed.

## 3. Results

### 3.1. System Set Up/Approach

When the first case of COVID-19 was reported in the United States, our system began preparing for the upcoming pandemic. The system-wide and HICS structure was based on the Federal Emergency Management Agency disaster manual [4,9]. Infection control specialists were assigned to the safety officer role due to the need for expert recommendations for prevention and detection [10]. Objectives were established on a daily basis and reviewed by the incident commanders who retained full authority over all hospital operations. An online, centralized dashboard was used to track and reallocate resources across the hospital system in real-time, which included all supplies ranging from IV-tubing to N95 masks to ventilators. In concert, a triage system was also finalized to coordinate patient admissions throughout our 10 hospital system and to ensure that patients who required ECMO were promptly transferred to our center. Triage decisions were governed using a multidisciplinary approach including a central intensivist physician (CIP), who focused on intensive care unit (ICU) admissions, and an access center physician (ACP), who maintained an overview of all hospital admissions and overall system capacity. Both made final decisions on patient destination and reviewed the needs of a system hospital (Figure 1). By creating a tiered ICU system, patients were efficiently transferred to the specific ICU in a designated hospital that best matched their clinical needs. Inter-institution transfer within the region utilized pre-existing private and public, ground and air transport systems. Over 2100 patients were successfully admitted to system hospitals; 29% of these patients required critical care. The BCU was initially the only ICU taking COVID-19 patients in the hospital, but eventually several other units of varying care intensity were dedicated for patients with COVID. Ultimately, a COVID-only system hospital was established, with a 55 ICU and 205 non-ICU patient capacity (Figure 2). 

During the 2014 Ebola crisis, a BCU was initially created as described by Smith et al. [11]. At that time, the unit only had two negative pressure rooms with specially trained staff. A communication system was also developed using secure video chats via tablets. While it was never used for Ebola patients, this served as the basis for the BCU during the COVID-19 pandemic. In 2020, the BCU became the designated unit for care of critically ill patients requiring ECMO support in the COVID-19 pandemic. Staff and resources were reallocated from a heterogeneous pool of critical care and stepdown units to admit and care for the most critically ill COVID-19 patients in the entire system. 

Lessons:Early planning is critical when facing a disaster event, such as a viral respiratory pandemic.Creation of a formal HICS structure with assigned triage officers is vital for preserving the workforce, abolishing redundant processes, and improving overall patient access throughout a healthcare system.A centralized, computerized platform to track resource availability can inform decisions regarding scarcity of supplies.

### 3.2. Evolution of the Biocontainment Unit

At first, the BCU had five individual negative-pressure rooms. As the census expanded, the unit was converted to a 16-room unit behind a single airlock. An airlock unit can be advantageous when caring for a large number of patients with the same contagion. It allows for the preservation of personal protective equipment (PPE), while also allowing for a swift response to emergent issues as many healthcare professionals are already donned within the unit [10]. Each room contained a tablet for communication and a telemetry monitor, both visible outside the airlock. Outside of the airlock, there were break rooms, a communication command room with a bank of tablets, and donning/doffing areas. As has been reported in Singapore, there was an emphasis on daily monitoring of symptoms and a dedicated facility for personal sanitation for providers after working in the BCU [12]. There was also a tranquility room with massage chairs and music, to alleviate the strain on mental health during a pandemic [13] (Figure 3). Similar to what had been seen in other centers, there was a small outbreak among the staff that was traced to non-adherence to social distancing [14]. As a response, strict social distancing rules were re-enforced, and additional break rooms were created to ensure no more than four people occupied any one room at a time.

While multiple units cared for COVID-19 patients, the BCU handled the most critically ill within both the hospital and the system, including all patients on VV-ECMO. As was common at many institutions, in order to accommodate the surge of patients, “double-bunking” was eventually required. Each room in the BCU could house one ECMO and one non-ECMO patient, the maximum concurrent number of VV-ECMO patients was capped at 16. Within a few weeks of our first cannulation, the number of VV-ECMO patients surpassed the capacity of the BCU. In order to continue accommodating additional transfers who may require VV-ECMO, some patients from the BCU were transferred to another ICU within the same building. These were generally patients displaying sustained clinical improvements and nearing the end of their VV-ECMO course. For a brief period, the medical center reached its ability to offer additional patients ECMO support. As a result, outside evaluations and transfers for VV-ECMO had to be declined and were referred to other regional centers. In efforts to decrease overall census, patients past the most critical portion of their course or those determined not to be candidates for VV-ECMO, were repatriated to their hospitals of origin. The average duration of VV-ECMO for this cohort was 40 ± 30 days. 

The challenge with a dedicated airlock unit that implemented double-bunking was the risk of cross-contamination. Instead of taking care of patients in individual rooms, where PPE is donned and doffed per room, PPE is donned and doffed when traversing the airlock. In this setting, it may be easy to forget to treat the donned PPE as a base-layer and follow additional universal precautions when moving between rooms as the psychological effect of already having PPE on makes these otherwise routine practices easy to overlook. When coupled with double-bunking, this has the potential to lead to cross-contamination between patients and requires heightened vigilance. The potential for spreading multidrug-resistant organisms, nosocomial infections, and other viral pathogens in an airlock with double-bunking can lead to significant morbidity and mortality. Eventually, as the first wave started to abate, and cognizant of the risk of cross-contamination with double-bunking, the BCU transitioned back to single rooms (Figure 4). 

Lessons:An airlock unit for treating a contagious pathogen can facilitate the care of a large group of critically ill patients.Social distancing of staff outside the airlock is crucial to minimizing the risk of secondary exposures.The added capacity with double-bunking critically ill patients comes with the potential for increased cross-contamination and should be considered a last resort, even in a pandemic.

### 3.3. Intensivist and Provider Teams

Per initial CDC reports, up to 11.5% of all patients with COVID-19 required ICU admission, which has caused a strain on many hospitals [15]. During the initial phase of the COVID-19 pandemic, we were able to maintain our traditional model of a single attending during the day with home call at night. As part of our standard staffing model, there were in-house critical care attendings available if needed. However, as the volume increased and double-bunking occurred, a tiered staff model was implemented (Figure 5A). In addition to the critical care team, there was also a dedicated team of cardiac surgeons available for cannulations as well as assistance in the decision-making and management of the ECMO patients.

Additional intensivists were chosen based on the Society of Critical Care Medicine (SCCM) recommendations [16]. The provider pool consisted of advanced practice providers and fellows with familiarity in taking care of ECMO patients. Initially, designated providers would go into the airlock, but as the BCU expanded it was no longer feasible for a single provider to carry out all the necessary tasks. It became necessary for multiple providers to go into the airlock throughout a shift. Patient care algorithms, template notes, and multiple daily reports were utilized as an attempt to standardize and maintain continuity of care. Despite this, as the patient census increased, there was also an increase in the variance of care as a result of differing levels of experience with surgical critical care and ECMO within the augmented attending and provider pool. Recognizing this, the intensivist assignment was modified to have dedicated attendings to treat ECMO and non-ECMO patients, similar to the experience in the LRU and previous ECMO specified teams (Figure 5B) [10,17]. 

Lessons:A tiered staffing model to employ team-based approach from attendings to providers to nurses is effective.A dedicated ECMO and non-ECMO intensive care team is beneficial when caring for a mixed unit.Maintaining a standardized approach for continuity of patient care is difficult but vital to the care of critically ill patients during a pandemic

### 3.4. Nursing

Since the initial plans for a BCU model in response to the Ebola Virus epidemic, a specialized 40-member team was formed with nurses, therapists, and patient care technicians from units throughout the institution. Nurse specialties on the team included trauma, surgical, medicine, and pediatric critical care, as well as those from intermediate care units. Staff continued to work in their home units until activated by HICS.

Once activated for the COVID-19 pandemic, it was quickly recognized that additional members would be required. A rapid onboarding of more than 70 nurses and staff joined the existing 40-member BCU team to meet the needs of the COVID-19 surge. A team nursing model was established that mixed specialties and competencies, and had both ICU and stepdown nurses, allowing for teams to take care of more patients [18]. There were a total of four teams for the 16 BCU rooms. In addition, float nurses, nurses dedicated for continuous renal replacement therapy, patient care technicians, and occupational therapists were present in the airlock. All nurses alternated working in the airlock throughout a shift. This rotation was also utilized by respiratory therapy (RT) and the ECMO specialists to keep a constant presence and availability in the BCU. 

Though this model increased the patient-to-nurse ratio without the dedicated critical-care-trained nurses, the departure from a 1:1 ratio did have drawbacks [19]. As team nursing was utilized to care for these patients, patients would not have as immediate or attentive treatment with change in clinical states [20]. Coupled with the effort to limit consumption of PPE and exposure, physical assessments were done by nursing and relayed over tablet communication and electronic medical charts. This model was challenging as the nurse reporting assessments during rounds may not have yet assessed the patient due to the rotation schedule for the shift, or was the only critical care nurse that was participating in a resuscitation. In light of this, as well as an increase in availability of PPE, providers increased frequency inside the airlock for first-hand exams.

Lessons:Team nursing can allow for increased patient-to-nurse ratios; however, there are trade-offs that must be considered.Understand the limitations of patient assessments when relying on team nursing, especially when reallocating staff from non-critical care settings, and the importance for firsthand provider assessments.

### 3.5. Communication/Technology

The care of patients relies on frequent and open communication within the healthcare team, as well as between the healthcare team and the patient or family. This is crucial for patients within the ICU, particularly those of critical acuity [21]. When compounded with staff reassignment, an airlock, strict no visitor policy, and resource scarcity during a pandemic, it poses unique challenges.

With staff reassignments, team members may be caring for a patient population previously unfamiliar to them. As moving in and out of the airlock requires donning and doffing PPE, expeditious in-person communication between team members across the airlock is severely hindered. Continuous video streaming on tablets in patient rooms became a mainstay of communication. A staff member in the communication command room had a view of video streams of all BCU patient rooms and could quickly facilitate communication between team members both within and across the airlock. This was further augmented with an already established Health Insurance Portability and Accountability Act compliant secure messaging application that was easily accessible on computers and mobile phones that remained within the airlock.

Early during the pandemic, the hospital adopted a strict no-visitor policy. Initially, providers and nurses caring for patients communicated to families with daily updates. If goals-of-care or end-of-life discussions occurred, a virtual meeting was arranged involving multiple team members. Communication was difficult to maintain as more patients were admitted to the BCU. With increased patient-to-care team ratios, this led to interruptions in necessary clinical workflow. A team was created consisting of palliative care, social work, and a dedicated BCU staff member to maintain communication with families. In addition, multiple times a week, online video streaming was used for families to see patients via the tablets already in the rooms. 

Over the course of the pandemic, the dramatic increase in use and subsequent scarcity of resources, such as PPE and therapeutic drugs, led to daily fluctuations in institutional availability. While on a system-level this information was meticulously tracked, the dissemination of this information to individuals on patient care teams was not as consistent. This led to perceived shortages, mainly of medications, that were propagated throughout the team by word-of-mouth. Circulating this information via constant email communication was utilized; however, it may get overlooked especially in the sea of daily emails. Daily morning reports with the medical director were also implemented. A centralized dashboard for system resources pertinent to the daily care of patients that is accessible by designated key members on the team, namely pharmacists, is useful.

Lessons:Expeditious communication within the team is essential for the care of critically ill patients.The use of technology (tablets, mobile phones, secure messaging applications) can prove helpful within an airlock unit.Consider the use of a designated team to communicate with family members for routine updates.A centralized system to track availability of medications that is accessible by the care team helps work around true versus perceived shortages.

### 3.6. ECMO Planning

Though the institutional capacity of approximately 800 beds did not change, we increased from 166 ICU beds pre-COVID up to 276 ICU beds at the peak of the initial surge in our institution, with the capability to further augment critical care capacity if needed. With the knowledge that critical illness in COVID-19 patients was primarily manifesting with respiratory decompensation, and reports of rapidly increasing ECMO use from other centers around the world, we sought to expand our ECMO capacity in concert with our increase in capacity for critical care. [22,23,24,25,26]. The first priority was stockpiling disposables (circuits, cannulae, and oxygenators) in early February. During the spike of cases in New York City, rental units were in short supply, so we sought other ways to expand our capacity.

At different stages throughout the pandemic, we were able to augment our ECMO capacity by reallocating ECMO units from other hospitals in our system, renting additional Rotaflow units (Getinge, Wayne, NJ, USA), purchasing Cardiohelp units (Getinge), adding an oxygenator to our in house CentriMag devices (Abbott, Abbott Park, IL, USA), as well as trialing Novalung (Xenios AG, Heilbronn, Germany) and Quantum (Spectrum Medical, Fort Mill, SC, USA) ECMO platforms. Through this, we went from 19 ECMO units to 37, and an average of 8–10 adults on ECMO pre-COVID-19 (~160–180 runs of ECMO/year) to a peak of 28 patients during the pandemic—20 of those being COVID-19 positive. With multiple ECMO platforms in-house, the conscious decision was made to only use our most abundant unit, Rotaflow, in the BCU to minimize variability.

Another challenge encountered was acquisition of ancillary ECMO equipment, most notably the heater-coolers and gas blenders. Rental units were not readily available, nor were units for purchase, thus we used other strategies until we were able to obtain more. Prior to cannulation, if patients were normothermic or hyperthermic, we decided to use forced-air warming blankets as the mode of temperature regulation in lieu of a heater-cooler. Some patients had persistent shivering requiring medical intervention. For patients that needed active warming, heater-coolers from our cardiopulmonary bypass circuits were utilized. Air regulators on the wall in patient rooms were utilized in the absence of additional blenders. Tubing was connected to the regulator, allowing 100% oxygen to be used for sweep gas flow in patients who were more stable and required lower sweep. It was important to be aware of the tubing and ensure that it did not inadvertently become disconnected as it traversed the room to the ECMO circuit.

Contracts were signed with two separate perfusion staffing companies to augment our staff. Ventricular assist device (VAD) engineers and RT staff were also trained and available if needed. In order to keep track of all the COVID-19 ECMO patients with the rotating staff, a large whiteboard with key clinical parameters (e.g., date of cannulation, flow, sweep, etc.) was placed in a central location outside of the airlock. While a majority of resources were going to the care of COVID-19 ECMO patients, we also had to maintain the care of our non-COVID-19 ECMO patients, such as the emergent adult and pediatric cardiac surgical patients. In a pandemic, these patients must be accounted for and play a role in the limitations and/or capacity of a hospital.

Lessons:With an anticipated increased need for ECMO, the process of acquiring additional equipment should begin early.Total ECMO capacity can be augmented by various means such as reallocating from within the system, adding an oxygenator to extracorporeal VADs, and trialing new platforms.Minimize variability when caring for high volumes during a pandemic.Have an algorithmic approach to augmenting ECMO staff and consider outside contracts as well as training VAD and RT staff.Consider the use of alternatives when there are shortages of ancillary ECMO equipment such as forced-air warming blankets for temperature management or room in-line air for sweep flow.Have a central place to update important clinical parameters for ECMO patients with rotating staff, intensivists, and cardiac surgeons.

### 3.7. Physical Therapy

The LRU, along with only a few select institutions, routinely ambulated ECMO patients [27,28]. Early mobilization and increased physical therapy lead to reduced cognitive and muscular deconditioning [29,30]. However, it requires a team of physical therapists, nurses, and patient care technicians for each patient, which requires dedicated resource allocation similar to provider and nursing teams. During the initial response phase, we were able to train therapy staff in the appropriate therapy delivery while maintaining PPE. In the midst of the surge, this intervention remained far below the standard in our ICU due to several limitations including sparse PPE, personnel, and space with double-bunking. This was augmented by Catalyst critical recovery beds (KREG Therapeutics, Melrose Park, IL) and nursing delivered therapy. Once patients were decannulated and transferred out of the BCU, they had more extensive physical therapy.

Lessons:Staff and PPE within an airlock unit may limit the ability to provide physical therapy and safely ambulate patients.A strict schedule of positioning changes, in the absence of physical therapy, is important for prevention of pressure injury.Consider the routine use of alternatives, such as positioning beds, in this setting.

## 4. Discussion

Preparing for a surge of critically ill patients during a pandemic can be a daunting task. The implementation of a coordinated, system-level approach can help with the allocation of resources as needed. Focusing on established strengths of hospitals within the system can guide triage based on individual patient needs. 

The use of ECMO in the care of patients with acute respiratory failure requires significant resources and expertise, both at a hospital-level and within the care team [31,32]. We have outlined our approach for the care of a large number of patients on ECMO. In a pandemic, other patients with ECMO needs must be accounted for. While an airlock unit has advantages in this situation, it also poses unique challenges. Though double-bunking may be unavoidable when accommodating surge-levels of patients, the potential for cross-contamination requires strict adherence to universal precautions. Double bunking should only be used if absolutely necessary. 

Increasing the capacity for ECMO requires significant financial and clinical commitment from the institution. Staffing increased four-fold compared to the previous dedicated LRU for the care of these patients, and roughly over $3 million of expenditure to reach our expanded capability. The management of ECMO patients is still a specialty care, and a systematic and hospital-based approach requiring an ECMO team composed of multiple experienced individuals is paramount during a respiratory viral pandemic. We have found that many of the lessons we have learned from our ECMO effort have been applicable to pandemic care in general and hope they can provide insight to other centers. 

## Figures and Tables

**Figure 1 membranes-11-00258-f001:**
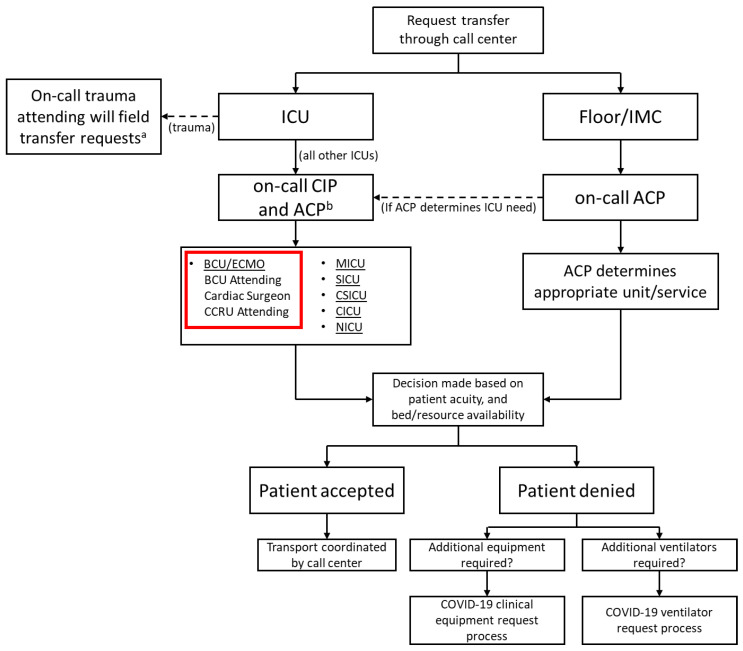
Institution triage system. ^a^ for any COVID-19 patients or patients requiring critical care, the CIP will be added to the cell center consult; ^b^ ICU need is identified by CIP with ACP assisting by providing a system-wide overview and identifying available resources Legend: ACP—access center physician, BCU—biocontainment unit, CCRU—critical care resuscitation unit, CICU—cardiac intensive care unit, CIP—central intensivist physician, CSICU—cardiac surgery intensive care unit, ECMO—extracorporeal membrane oxygenation, ICU—intensive care unit, IMC—intermediate care, MICU—medical intensive care unit, NICU—neurology intensive care unit, SICU—surgical intensive care unit.

**Figure 2 membranes-11-00258-f002:**
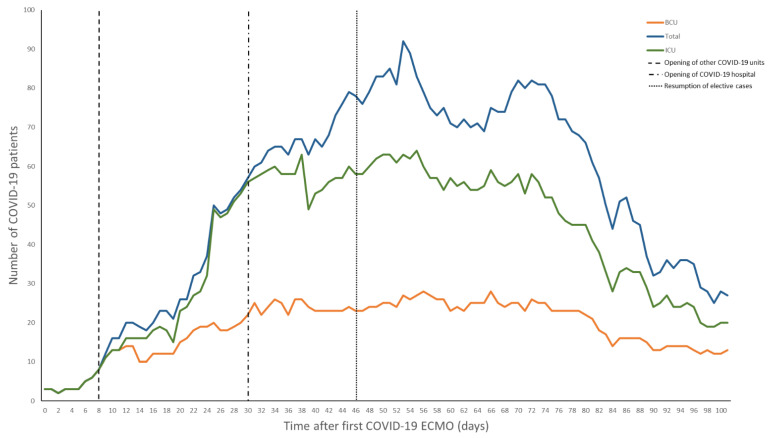
Positive patients admitted to our institution. Day 0 is when the first COVID-19 patient was placed on extracorporeal membrane oxygenation (ECMO). Legend: BCU—biocontainment unit, ECMO—extracorporeal membrane oxygenation, ICU—intensive care unit.

**Figure 3 membranes-11-00258-f003:**
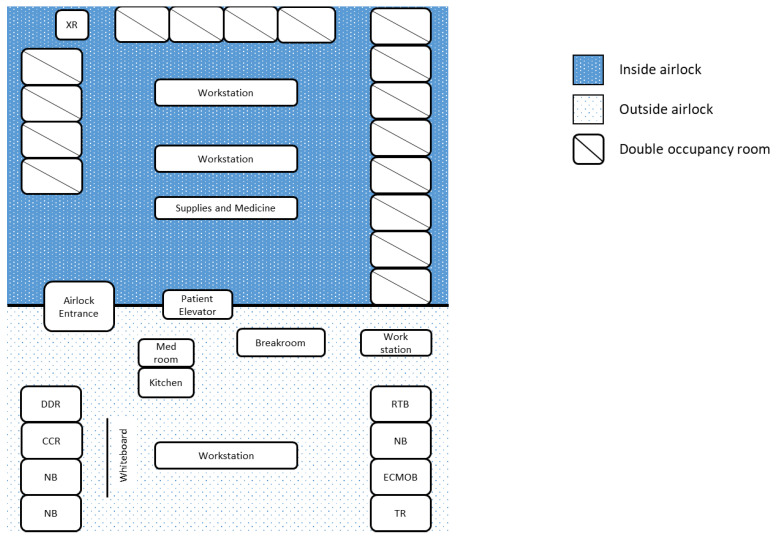
COVID-19 biocontainment unit schematic. Legend: DDR—donning and doffing room, ECMOB—ECMO specialist breakroom, CCR—communication command room, NB—nurses’ breakroom, RTB—respiratory therapist breakroom, TR—tranquility room, XR—portable X-ray machine.

**Figure 4 membranes-11-00258-f004:**
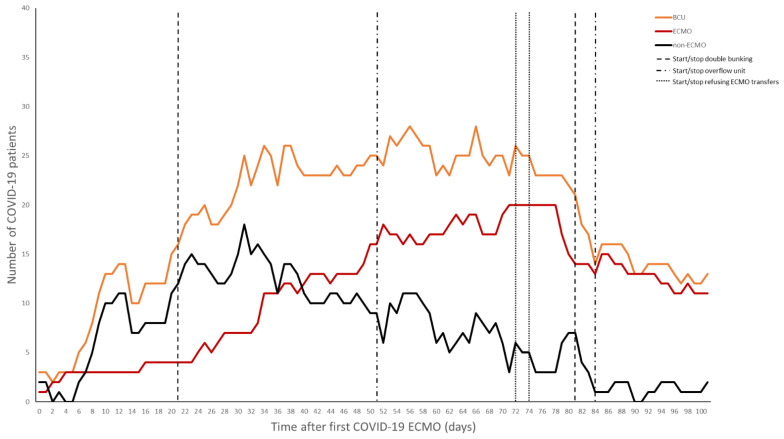
Total patients in the biocontainment unit and number of ECMO versus non-ECMO patients over time.

**Figure 5 membranes-11-00258-f005:**
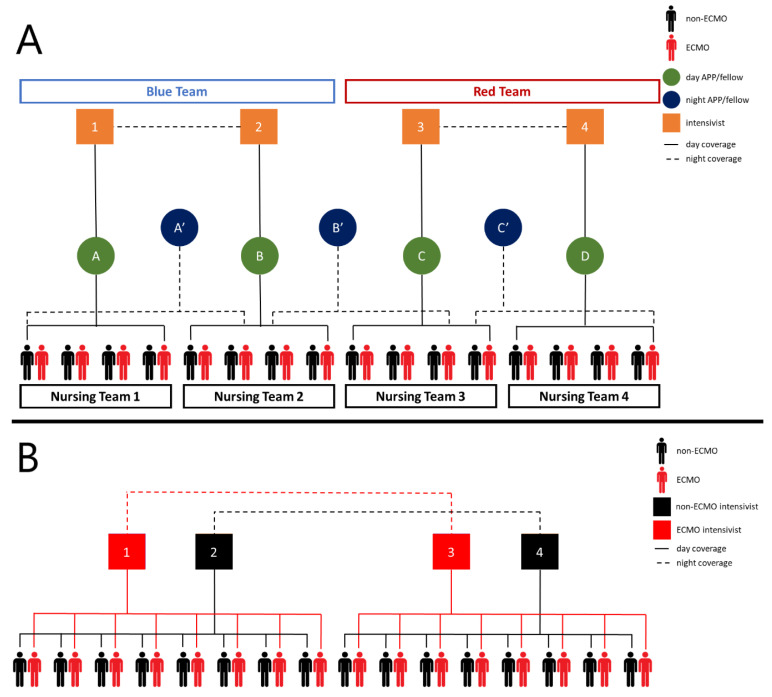
(**A**) Initial intensivist/(**B**) provider care team schematic. The biocontainment unit (BCU) is split into two teams. During the day, each intensivist oversees an advanced practice provider/fellow, and overnight one intensivist from each Table 24 h shifts. In regards to nursing, the BCU was split into four groups and one nursing teams cared for all of the patients in that group. (**B**) Revised intensivist/provider care team schematic while double bunking. The unit is split in half. Intensivist assignment changed and the rest remained the same. An ECMO and non-ECMO cover ECMO and non-ECMO patients, respectively, on their half of the unit during the day. Overnight, one ECMO intensivist covers all ECMO patients and one non-ECMO intensivist covers all non-ECMO patients. Intensivists 1 and 3 alternate between day shifts and 24 h coverage as do intensivists 2 and 4. Legend: APP—advanced practice provider.

## Data Availability

The datasets generated and/or analyzed during the current study are available from the corresponding author on reasonable request.

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
