# Peer review of "A Dedicated Veno-Venous Extracorporeal Membrane Oxygenation Unit during a Respiratory Pandemic: Lessons Learned from COVID-19 Part I: System Planning and Care Teams"

_membranes, 2021, doi:10.3390/membranes11040258_

Round 1

Reviewer 1 Report

Thank you for giving me the opportunity to revise this excellent manuscript on hospital prepadness in the COVID 19 first wave and challenges to provide VV ECMO.

I have only minor comments that the reader would appreciate

  • Average duration of VV ECMO support
  • Number of pats which were refused or not accepted due to the lack of resources
  • Attention: line 184-86 is redundant.

Reviewer 2 Report

This manuscript describes the planning of care and teams in surge of the Covid 19 pandemic. This detailed description provides insight and lessons learned for other centers worldwide. I just have a few questions for the authors:

Could you state in the introduction your view on the use of ECMO in this pandemic? is it justifiable to provide such a complex treatment for a small number of patients in light of the gigantic number of critically ill patients. Is there a risk of denying patients critical care because ECMO is used.

Could you describe the hospital in more detail? how many beds, ICU beds, normal ECMO runs/year.

Is this planning for ECMO patients alone or could most of these lessons apply to covid pandemic care in general?

Was there a statewide/regional committee distributing covid patients?

Please provide some detail on transporting in your region.

The section on physical therapy seems out of place in this article. this might belong in an article describing clinical care.
